# Assessment of Nit-Occlud atrial septal defect occluder device healing process using micro-computed tomography imaging

Elodie Perdreau[1,2,3]*, Zakaria Jalal[1,2,3,4], Richard D. Walton[1,2,3], Matthias Sigler[5], Hubert Cochet[1,2,3,6], Jérôme Naulin[1,2,3], Bruno Quesson[1,2,3], Olivier Bernus[1,2,3], Jean-Benoît Thambo[1,2,3,4]

1 Electrophysiology and Heart Modeling Institute, IHU Liryc, Fondation Bordeaux Université, Pessac-Bordeaux, France, 2 U1045, Centre de recherche Cardio-Thoracique de Bordeaux, Université de Bordeaux, Bordeaux, France, 3 U1045, INSERM, Centre de recherche Cardio-Thoracique de Bordeaux, Bordeaux, France, 4 Congenital and Pediatric Cardiology Unit, Bordeaux University Hospital, Pessac, France, 5 Pediatric Cardiology and Intensive Care Medicine, Georg-August University Hospital, Göttingen, Germany, 6 Cardiothoracic Pole, Bordeaux University Hospital, Pessac, France

* elodie.perdreau@chu-lyon.fr

**Data Availability Statement:** All relevant data are within the paper and its Supporting Information files.

## Abstract

After percutaneous implantation of a cardiac occluder, a complex healing process leads to the device coverage within several months. An incomplete device coverage increases the risk of device related complications such as thrombosis or endocarditis. We aimed to assess the device coverage process of atrial septal defect (ASD) occluders in a chronic sheep model using micro-computed tomography (micro-CT). After percutaneous creation of an ASD, 8 ewes were implanted with a 16-mm Nit-Occlud ASD-R occluder (PFM medical, Cologne, Germany) and were followed for 1 month (N = 3) and 3 months (N = 5). After heart explant, the device coverage was assessed using micro-CT (resolution of 41.7 μm) and was compared to histological analysis. The micro-CT image reconstruction was performed in 2D and 3D allowing measurement of the coverage thickness and surface for each device. Macroscopic assessment of devices showed that the coverage was complete for the left-side disk in all cases. Yet incomplete coverage of the right-side disk was observed in 5 of the 8 cases. 2D and 3D micro-CT analysis allowed an accurate evaluation of device coverage of each disk and was overall well correlated to histology sections. Surface calculation from micro-CT images of the 8 cases showed that the median surface of coverage was 93±8% for the left-side disk and 55±31% for the right-side disk. The assessment of tissue reactions, including endothelialisation, after implantation of an ASD occluder can rely on in vitro micro-CT analysis. The translation to clinical practice is challenging but the potential for individual follow-up is shown, to avoid thrombotic or infective complications.

## Introduction

Intra-cardiac percutaneous devices are now the treatment of choice for the closure of congenital heart defects such as atrial septal defect (ASD) or ventricular septal defect (VSD) as well as

**Funding:** This study received financial supports from the French Government as part of the "Investments of the Future" program managed by the National Research Agency (ANR), Grant reference ANR-10-IAHU-04 and the grant "Aide à la recherche par équipe 2018, Cardiopathies de l'enfant" from the French Federation of Cardiology.

**Competing interests:** The authors have declared that no competing interests exist.

**Abbreviations:** ASD, atrial septal defect; CT, computer tomography; LAA, left atrial appendage; MRI, magnetic resonance imaging; PBS, phosphate buffered saline; PFO, patent foramen ovale; VSD, ventricular septal defect.

for the closure of the patent foramen ovale (PFO) and the left atrial appendage (LAA) to prevent thromboembolic complications [1–3].

The healing process following the implantation of such devices is of high importance. Knowing the course of this process aids in the determination of the choice of treatment and its duration, to prevent device related complications such as thrombus formation or infective endocarditis.

Both animal models and explanted devices from humans have been used to study the healing process. Initially, an intense inflammatory reaction is observed involving fibrin, inflammatory cells, angiogenic factors, extra-cellular matrix and thrombotic material; it is then followed by the formation of a granulating tissue evolving towards fibrosis [4–6]. The duration of this process is highly variable, from 28 to 90 days in animal models [7,8] and from 6 to beyond 18 months in humans [9].

Delayed cases of device-related thrombotic and infectious complications have been reported, sometimes years after implantation [10–15]; associated with an incomplete device coverage. Hence, despite lengthy healing processes, complete covering of devices may sometimes not be achieved, while no individual predictive criteria have been identified.

Assessing the individual coverage of an intra-cardiac device therefore appears essential to adapt the post-implantation preventive treatment. Non-invasive imaging techniques such as computed tomography (CT), enables high resolution imaging of both tissue and occluder components while considering their three-dimensional configuration. The choice of ASD occluder device stems from it being the longest serving implanted intra-cardiac device, with a previously demonstrated high reliability in animal models.

In this preliminary work, we aimed to assess the device coverage process of ASD occluder devices in a chronic sheep model using computed tomography. This process was analysed as follows: identification of the covering tissue in 2D and 3D, calculation of the surface area of coverage and mapping of the thickness of the tissue coverage.

## Methods

### Percutaneous device characteristics

The Nit-Occlud ASD-R occluder (PFM medical, Cologne, Germany) is a recently developed ASD occluder device. It consists of two disks manufactured from a single nitinol wire (thickness 170 microns) forming a mesh and two polyester sulfone membranes, used to promote endothelialization. Both disks are attached to each other with a narrow connecting waist. There is no protruding fixation-clamps. The polyester membrane resides underneath the right metal disk whereas left atrial disk is covered on the exterior surface. A reduced metal content and absence of protruding fixation clamps improved the compatibility of the Nit-Occlud ASD-R occluder with computed tomography-based imaging techniques, which are prone to artefacts associated with over attenuation of X-rays.

### Animal model

Eight cast ewes (mean age 5.4 ± 0.7 years old; mean weight 55.6 ± 7.9 kg) were enrolled for this experimental protocol, approved by the Ethical Committee of Bordeaux CEEA 50 (reference number: APAFIS#15508-201806140929827v2), in accordance with European legislation concerning animal experimentation (2010/63/UE; 2010). All procedures were performed under anesthesia and all efforts were made to minimize suffering. Animals were premedicated with ketamine (10–20 mg/kg), acepromazine (0.1 mg/kg) and buprenorphine (1.5 mL/kg). They were then anesthetized with intravenous propofol bolus (1 mg/kg), intubated and mechanically ventilated with isoflurane (1.5–3%) and oxygen/room air to maintain anesthesia. Arterial

line and gastric tube were fitted to respectively monitor arterial pressure during the procedure and collect gastric effluents. ECG, heart rate, temperature, respiratory rate and transcutaneous oxygen saturation, tidal volume and exhaled carbon dioxide were monitored throughout the procedure. The animal was warmed throughout the anesthesia using a 3M blower. Pain was released by using buprenorphine (10 μg/kg) and flunixine (2 mg/kg). A femoral venous access was obtained, and atrial trans-septal puncture was performed (Brockenbrough Needle, Medtronic, Inc. Mounds View, Minnesota) under fluoroscopic and transthoracic echocardiographic guidance (Probe 5S, Vivid S70, General Electrics) (Fig 1). Subsequently, dilation of the septal defect using a $16 \times 30$ mm Tyshak Balloon (NuMed, NY, USA), inflated at 2 atm was performed with success in all procedures. Heparin was injected to prevent thrombus formation (100 UI/kg). A 16-mm Nit-Occlud ASD-R occluder (PFM medical, Germany) was then deployed. After careful assessment of good device positioning and complete defect closure by transthoracic echocardiography, the device was released. Post-procedure management consisted in extubation after obtention of spontaneous breathing, analgesic support, with anti-inflammatory drugs, and antibiotics by amoxicillin (15 mg/kg) for eight days. The animal was placed in the initial housing conditions with appropriate refinement methods to its species: social group, salt stone available, hay and straw. The ewes were followed for ten days after the procedure by the veterinary team of the laboratory, with a daily clinical assessment (behaviour, locomotion, respiratory state, cardiac assessment) and pain management (Flunixine 2 mg/kg/d for 3 days, Buprenorphine if necessary). They were then sent to a housing for care until the end of the follow-up, the veterinary team intervening in case of complication. Acetylsalicylic acid was administered (200 mg/d) intramuscularly from implantation to euthanasia, as an antiplatelet therapy, based on human clinical practice. No specific sides effects were detected.

## Experimental protocol

After implantation, the device was explanted after 1 month (n = 3) and 3 months (n = 5), respectively (Fig 2). Then, they were euthanized using sodium phenobarbital after an injection of heparin to avoid clot formation on the device and sternotomy was performed to remove the heart. A precautious dissection of the heart was done to keep only the atrial septum with the device and parts of both atria (superior and inferior vena cava, pulmonary veins) to orientate the sample. After briefly flushing with saline, macroscopic evaluation and documentation was accomplished before fixation in formalin (buffered 10%).

## Micro-CT

A micro-CT scan of the block containing the device was then performed. Micro-CT is a 3D imaging technique using X-ray as standard CT scan imaging with a much greater resolution, with pixel sizes of 4–5 microns. Plastic markers were placed on the right disk: one close to the superior vena cava, one close to the inferior vena cava and the third one close to the septal leaflet of the tricuspid valve, to aid sample orientation.

To enhance contrast and increase image quality, the sample was placed in a solution containing an iodine-based contrast (Xenetics®) and phosphate buffered saline (PBS) solution for 24 hours. Before performing the scan, three successive solutions of PBS were used to wash the block that was then placed in a box without any liquid.

The micro-CT (SkyScan 1276, Bruker, Belgium) was configured for acquisition as follows: X-ray source of current 200 μA and voltage 100 kV were filtered by sheets of aluminium and copper; Images of dimensions 1008 x 672 were acquired at a pixel resolution of 41.7 μm; acquisition exposition times were 220–255 ms over rotation angles of 0.3˚ covering a 360˚ rotation and averaged over 20 frames. The minimal absorption value obtained was 10–15% and the

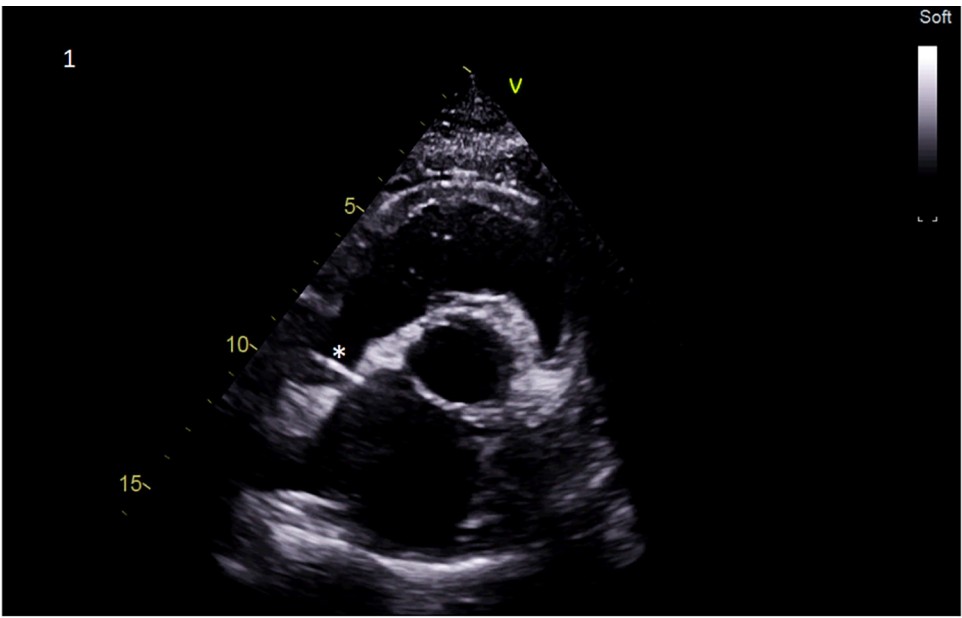

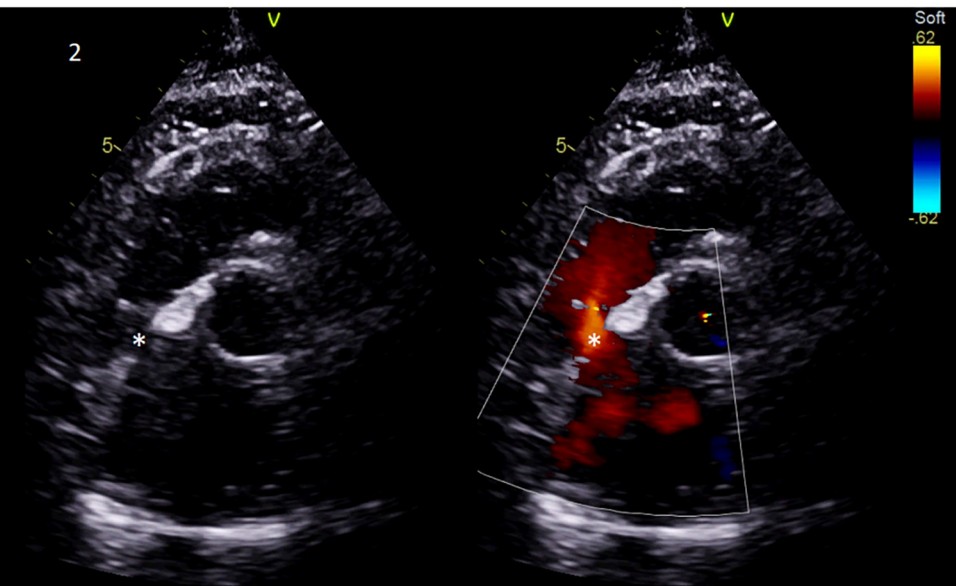

**Fig 1. Brockenbrough technique to create an ASD.** * Brockenbrough needle on picture 1 for atrial trans-septal puncture. * After the creation of an ASD shunt on picture 2, the red color stands for the left-to-right shunt through the ASD.

Sheep model (N=8)

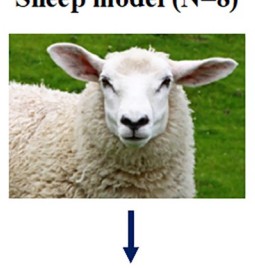

Atrial trans-septal puncture with dilation

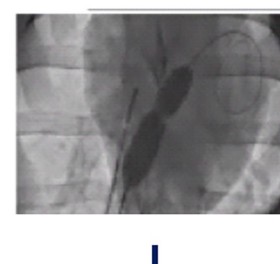

Implantation of a 16-mm Nit-Occlud ASD-R occluder (PFM medical, Germany)

Explantation with fixation in formalin

At 1 month (N=3)

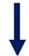

At 3 months (N=5)

Micro-CT

Histology

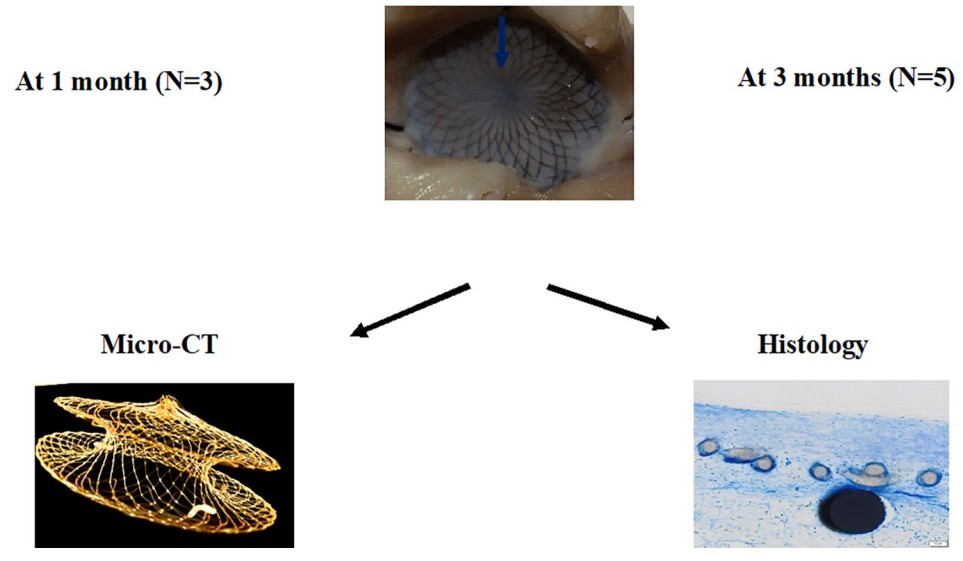

**Fig 2. Flow chart of the study.**

mean absorption was 63%. The duration of acquisition was seven hours for all animals, determined by the selected resolution of 41.7 μm.

## 3D reconstruction and visualization of device coverage

To prepare 3D reconstruction, X-ray projection images were corrected for misalignment (<1˚) and beam hardening (25%). No ring artefact reduction was necessary. Tomographic reconstructions with final isotropic resolutions of 41.7 μm were outputted in 16-bit TIFF format. Three-dimensional image smoothing, segmentation and visualization processes used Amira® software. To compare micro-CT and histology data, virtual sections passing through the inferior vena cava marker and the center of the device were obtained using the multiplanar visualization tool.

## Quantification of device surface coverage

An imaging processing pipeline was developed to quantify the percentage coverage of the occluder disks (Fig 3). The pipeline was formed from four main steps: (i) Segmentation of the occluder device; (ii) estimation of a solid disk surface geometry; (iii) segmentation of covering tissue and (iv) quantification.

**Segmentation of the occluder device.** Occluders were transformed to align their principal axis with the z-axis of the image volume. An initial segmentation using the lasso segmentation and interpolation functions in Amira enabled manual segmentation of the broad volume of interest including the occluder and coverage tissue yet removing the support and container.

**Estimation of a solid disk surface geometry.** The strongly X-ray attenuating nitinol wire of the occluder was segmented using a voxel intensity threshold. A Chamfer distance map was applied to background voxels from a binary image of the segmented nitinol wire, measuring the minimum distance of each voxel to the occluder material. A second binary image was derived using an upper threshold of 2.919 mm (70 voxels) on the distance map, to conserve distance values within proximity of the occluder and filling all spaces within its mesh structure. After which, an erosion and closing regime was used to reduce the second binary image to the original thickness of the nitinol wire thread, but conserving material filling the spaces in between the splines of the occluder.

**Segmentation of covering tissue.** An inverted binary image of the filled occluder was used to remove the occluder from the tissue-occluder greyscale image. A lower intensity threshold (15000 AU) was then applied to remove remaining background constituents and to derive a binary image of the covering tissue.

**Quantification of covering tissue thickness and surface area.** The occluder and covering tissue binary images were split in the XY plane to separate left and right disks into independent image volumes. A homemade Matlab routine was developed to quantify the tissue covering the outer occluder surfaces as follows: by iterating through each voxel in the XY plane, the Z axis was assessed in a direction starting from the outside of the device (in the atrial cavity). The number of voxels containing tissue prior to reaching the occluder surface were counted, converted to thickness using the image spatial resolution factor, and assigned to the corresponding xy pixel coordinates in a map of dimensions XY. A second map of equal dimensions for tissue coverage plotted values of 1 in locations where the occluder was observed prior to detecting tissue, and a value of 2 if tissue was observed followed by the occluder material. Note that a zero value was assigned for instances where tissue was observed in the absence of the occluder or for background. Coverage of the occluder was calculated as the percentage of tissue covering pixels in all non-zero pixels.

**1. Segmentation of the occluder device**

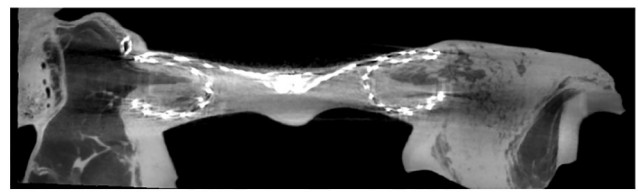

**2. Estimation of a solid disk surface geometry**

Segmentation of the nitinol wire

Chamfer distance map, erosion and closing regime

Isolation of the signal of the nitinol wire

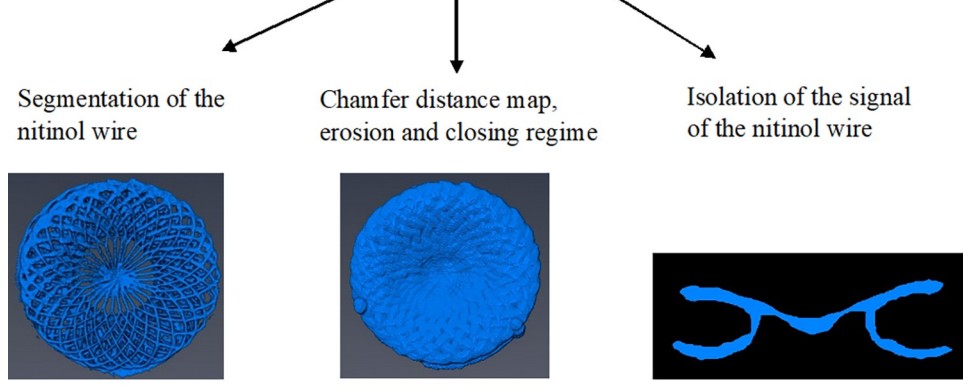

**3. Segmentation of the covering tissue**

Inverted binary image of the filled occluder applied to the volume

Obtention of a binary image of the covering tissue

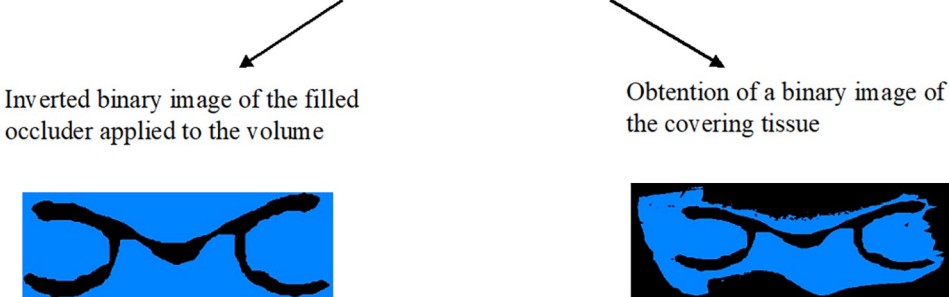

**4. Quantification of the tissue covering the outer occluder surfaces**

**Fig 3. Pipeline of the 3D reconstruction.**

**Histology.** After performing micro-CT scanning, all specimens were fixed in formalin after flushing with normal saline. The histopathological work-up followed a predefined protocol [5] with embedding of the specimen in a synthetic resin (methylmethacrylate; Technovit 9100, Kulzer&Co, Wehrheim, Germany). After hardening of the tissue blocks, sections of 0.8 mm were sawed using a diamond cutter (300CP, Exakt GmbH, Norderstedt, Germany). These sections were ground down to a thickness of 15–25 μm using a rotational grinder (400CS, Exakt GmbH, Norderstedt). Standard staining was performed with Richardson blue, that allowed to identify cellular and extracellular matrix components in blue, as previously described [5,6,16]. Histological slides were scanned with an automatic virtual microscope

scanning system for data storage and for enabling of virtual microscopy and measurements of tissue thickness (Olympus AS110 Fluorescence Virtual Slide Microscope Scanner, Olympus Europa Holding GmbH, Hamburg, Germany). The histologist was blinded for the results of micro-CT.

## Statistical analysis

Mean, median and standard deviation were used for quantitative data.

Assuming non-normal distribution of the data, non-parametric statistical tests were used.

## Results

### Device implantation procedure

After creating an ASD by transseptal puncture and balloon dilatation, a 16-mm Nit-Occlud ASD-R occluder (PFM medical, Germany) was successfully implanted in 100% of animals. Mean fluoroscopy and procedural durations were 13 ± 12 and 60 ± 35 minutes, respectively. There was no residual shunt at the end of the procedure. A cardiac tamponade occurred during the first procedure, due to left atrial appendage perforation by a guidewire, and was rapidly and successfully resolved by percutaneous drainage. No complication occurred during animal follow-up. Before animal sacrifice, transthoracic echocardiography showed no residual shunt in implanted devices.

### Macroscopic examination of explanted devices

After 1 month of follow-up, all the devices were covered with a thin layer of newly formed white connective tissue. This coverage appeared almost complete on the left atrial side of all the devices. For the right atrial disk, the coverage was incomplete and preferentially located on the borders of the devices.

After 3 months of follow-up, the left atrial side disk was completely covered with connective tissue, which had a greater thickness than that observed on the right disk (Fig 4). Right atrial side disk coverage was macroscopically complete in all but two devices. No wire strut fracture or thrombus formation was detected on the surface of the devices.

### Micro-CT

**Identification of the covering tissue in 2D and 3D.** Identification of the tissue was possible on 3D rendered reconstruction of raw images from computed tomography. Selective isolation of the covering tissue from the myocardium was possible as both tissues do not have the same intensity. A measurement of the thickness of coverage in regions of interest was done. Fig 5 presents the 3D rendered image after background-removed reconstruction of the disks of two devices.

**Comparison of micro-CT images and histology.** To validate findings observed by micro-CT, histology was performed on longitudinal cross-sections of all embedded occluder devices.

Histological sections allowed to identify the 3 layers of the covering tissue developed on the device: a thin outer neo-endothelium layer of less than 5 microns; a pseudo-intima layer with fibroblasts, extracellular matrix material and fibrin and a neo-tissue layer, in contact with metal struts and polyester membrane. There was no evidence of thrombus material.

The Richardson blue staining allowed to identify the tissue in blue and the metal struts in black, the polyester was also visible. Definite identification of endothelial cells based on morphological criteria was also obtained.

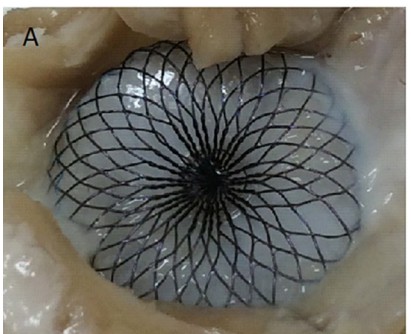
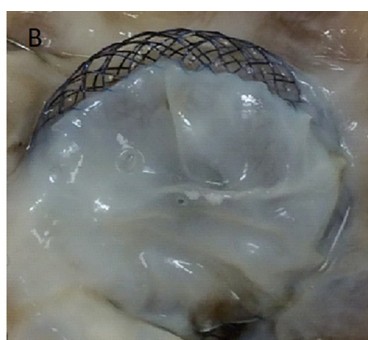

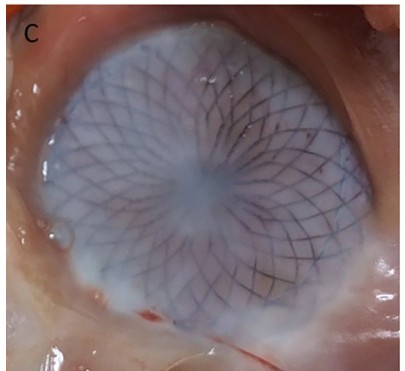
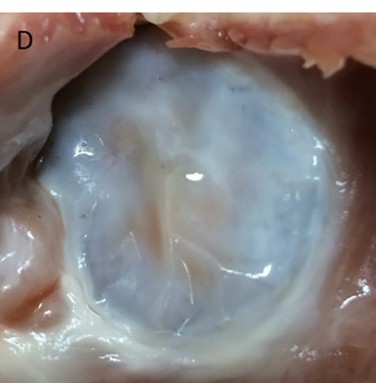

**Fig 4. Macroscopic examination of the disks of two devices (devices 5 and 8), both explanted at 3 months.** (A) Right-sided disk of device n˚5, with coverage located only on the edges of the disk. (B) Left-sided disk of device n˚5, with good coverage. (C) Right-sided disk of device n˚8 with complete coverage. (D) Left-sided disk of device n˚8 with complete coverage.

Histological images are shown in Fig 6 against micro-CT image slices extracted from the same approximate positions and orientations. A good visual correlation was noted, confirming the possibility to identify tissue coverage of this ASD occluder (Fig 7).

## Calculation of the surface of coverage

The results of measurement of the surface of tissue coverage of the left and right disks are presented as three-tone images in Fig 8. The tissue-covered occlude regions appear in white and uncovered occlude regions in grey. This was comparable to macroscopic examination data. The left disk was well covered, it was more variable for the right one with the three last implanted devices being almost totally covered as compared to the first five devices. Surface calculation from micro-CT images of the 8 cases showed that the median of coverage was 55 ±31% for the right-side disk and 93±8% for the left-side disk. After one-month follow-up, the median surface of coverage of the right-side disk was 41±17% and 86±6% for the left-side disk. After three-months follow-up, the surface of coverage was further increased on the right-side and the left-side disks (98±31% and 100±8%).

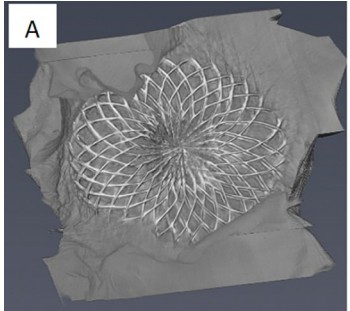
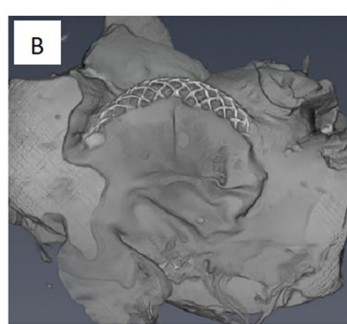

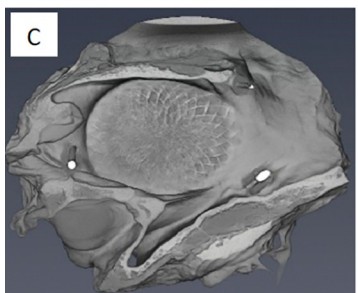
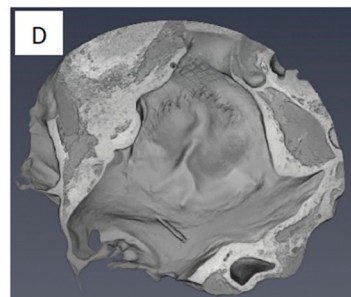

**Fig 5. 3D rendered image after background-removed 3D reconstruction of the disks of two devices (devices 5 and 8), both explanted at 3 months.** (A) Right-sided disk of device n°5. (B) Left-sided disk of device n°5. (C) Right-sided disk of device n°8. (D) Left-sided disk of device n°8.

## Mapping of the thickness of the coverage

The results of the mapping of thickness of the tissue coverage are presented in Fig 9. The thicker parts are red and the finer are blue. We can see great differences in thickness of the covering tissue, correlated to macroscopic assessment and CT scan data. The coverage of left-sided disk was almost complete in all devices with a thickness more likely to be important in the center of the device. As for the right-sided disk, the coverage was only present on the outline for five devices (1 to 5).

## Discussion

To assess tissue reactions after implantation of an ASD occluder device, micro-CT is a promising tool. We observed the possibility to evaluate the presence and the extension of tissular coverage on intra-cardiac devices (surface of coverage and thickness of the tissue), while preserving the 3D structure of the study object.

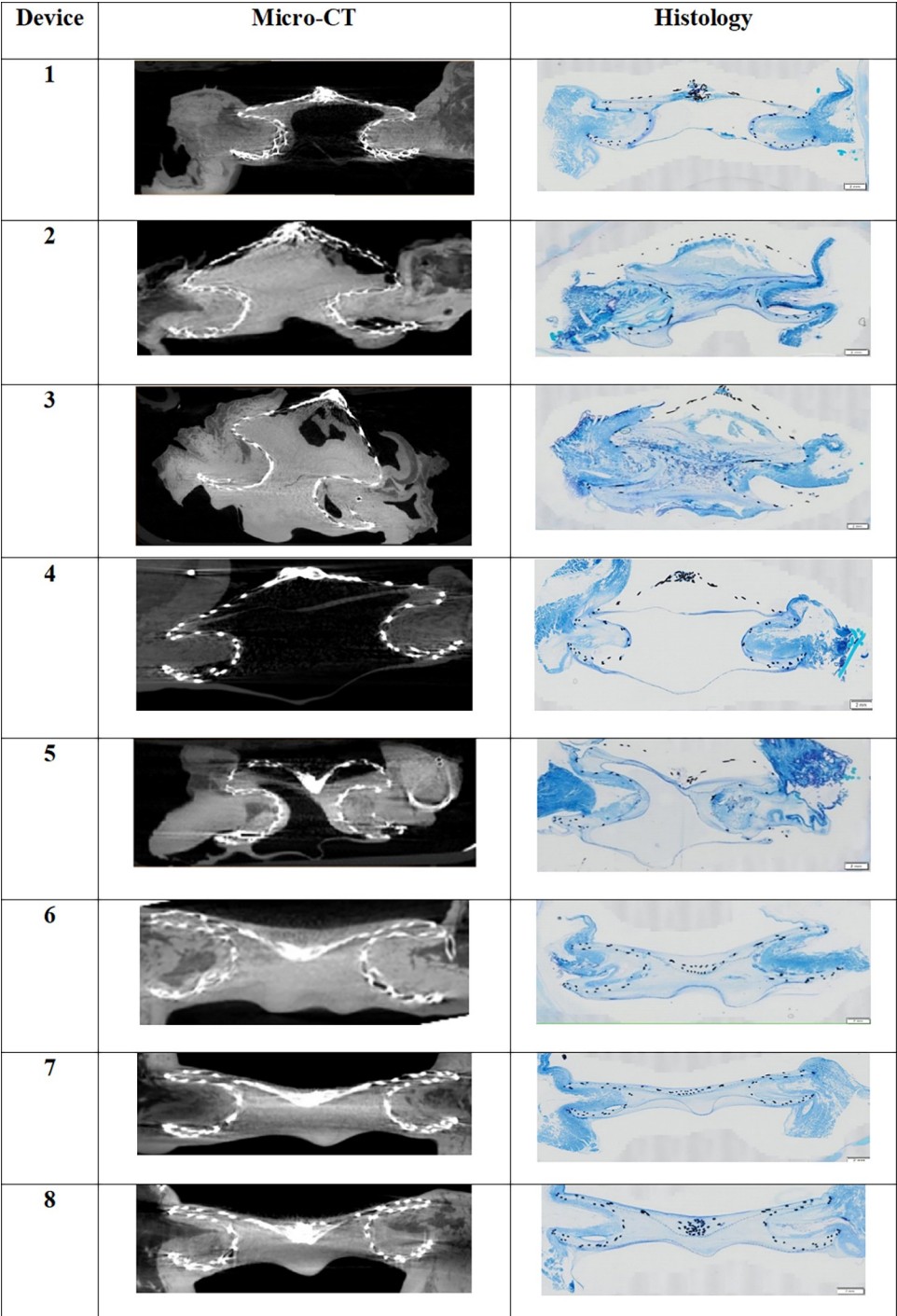

**Fig 6. Comparison of micro-CT and histology sections.** Devices 1–3 explanted after one-month follow-up. Devices 4–8 explanted after 3-month follow-up.

## Relevance of the results

The histologic description of the healing process related to ASD occluder implantation has been described previously through histological studies [5,6] and with strong concordance to

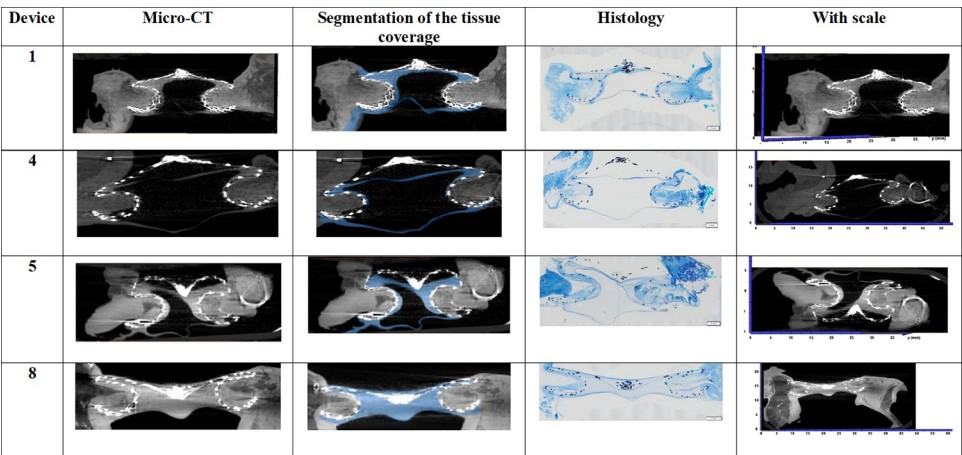

**Fig 7. Central Illustration presenting examples of segmentation of the tissue coverage to improve comparison of micro-CT and histology sections.** Device 1 explanted after one-month follow-up; Devices 4–8 explanted after 3-month follow-up. The first column shows a slice from raw 3D reconstruction of 4 devices presenting the occluder with covering tissue surrounded by myocardium. The secund column is a segmentation of the tissue that was formed in and around the disks of the devices. The third column shows the corresponding histology section of each device. The fourth column shows the slice from the first column with a scale.

the processes observed in sheep in our study (Figs 6 and 7). Yet histological assessment is limited to 2D cross-sections covering only a portion of the device, allowing for potential of misinterpretation of the broader coverage of the occluder.

We have shown that coverage can be incomplete but also heterogeneous throughout each surface of the occluder, particularly over short-term inspections of 1 month. Disk conformation appears to determine, in part, device coverage. The complete deployment of the occluder showed improved coverage but also a higher propensity for total transoccluder tissue replacement. Complete transoccluder tissue replacement in fully deployed occluders appeared to facilitate tissue coverage of the right disk. The mechanisms promoting elevated tissue replacement remains to be investigated. These may include the mechanical status of the surrounding tissue, smaller occlusion diameters, reduced transoccluder distances and the efficiency of septal occlusion.

## Micro-CT results

Identifying the tissue coverage was feasible and precise with micro-CT. This high-resolution imaging technique is still emerging in the cardiac field but has shown relevant applications in mapping valve and coronary calcification [17–20], to simulate coronary stent expansion [21–23] and to study cardiac and valve morphology [24–27]. Although *in-vivo* applications are limited by the high dose of radiation, the development of contrast agents (iodine agents, solutions of tungsten or barium ions) has enabled a reduction of exposure time [28,29].

Combined with histology and electronic microscopy, it allows the preservation of the 3D microstructure and precision of tissue analysis [28]. Micro-CT and histology appear complementary but not always comparable, as for the measurements based on these two techniques: the comparison is limited as the process of dehydration used before histology reduces the studied sample size and risks deformation of the sample. Concerning micro-CT, over attenuation of X-rays by the metallic occluder gives rise to over estimation of the diameter of the metallic splines. This image artefact can contribute to underestimation of the tissue coverage. Despite this, binary maps of tissue coverage were consistent with histological analysis.

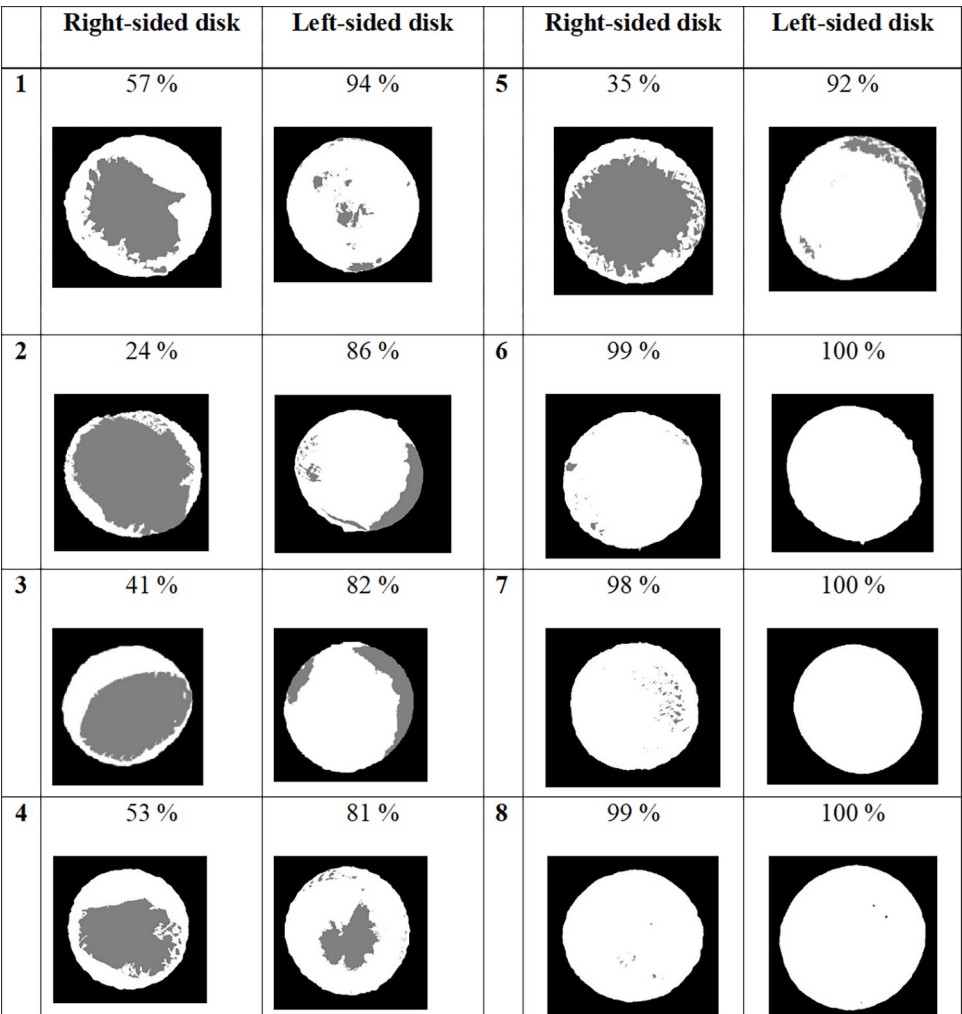

| | Right-sided disk | Left-sided disk | | Right-sided disk | Left-sided disk |
|---|---|---|---|---|---|
| 1 | 57 % | 94 % | 5 | 35 % | 92 % |
| 2 | 24 % | 86 % | 6 | 99 % | 100 % |
| 3 | 41 % | 82 % | 7 | 98 % | 100 % |
| 4 | 53 % | 81 % | 8 | 99 % | 100 % |

**Fig 8. Surface coverage of the disks of the eight devices.** Devices 1–3 explanted after one-month follow-up. Devices 4–8 explanted after 3-month follow-up. The surface of the disk which is covered with tissue appears in white.

## Potential clinical translation

The left atrial side disk coverage complete or near-complete coverage for all devices tested, whereas the right atrial side disk coverage was variable. It was observed in those occluders with low coverage that deployment of the occluder was incomplete and convex in shape, protruding into the right atrial cavity. In clinical practice, after percutaneous implantation of such devices, the follow-up of patients is typically performed by echocardiography. But this is without assessment of the device coverage and position of the device regarding the adjacent structures. Two recent studies offered an evaluation by CT-scan [30] and MRI [31]. Marini et al., showed that multi-slice computed tomography revealed, two years after the device implantation, moderate occluder protrusion in the region of the pulmonary veins and systemic veins, with a dynamic effect on 5 patients over 142 and a poor positioning in 2 of them [30]. Lapierre et al., used MRI to evaluate the relation of the device with adjacent structures during ageing, after the closure of a large ASD [31]. It showed a decrease in interactions of the device with adjacent structures, probably decreasing the risk of long-term complications [31]. These studies did not find major deformation of the disks and it was impossible to evaluate the device coverage.

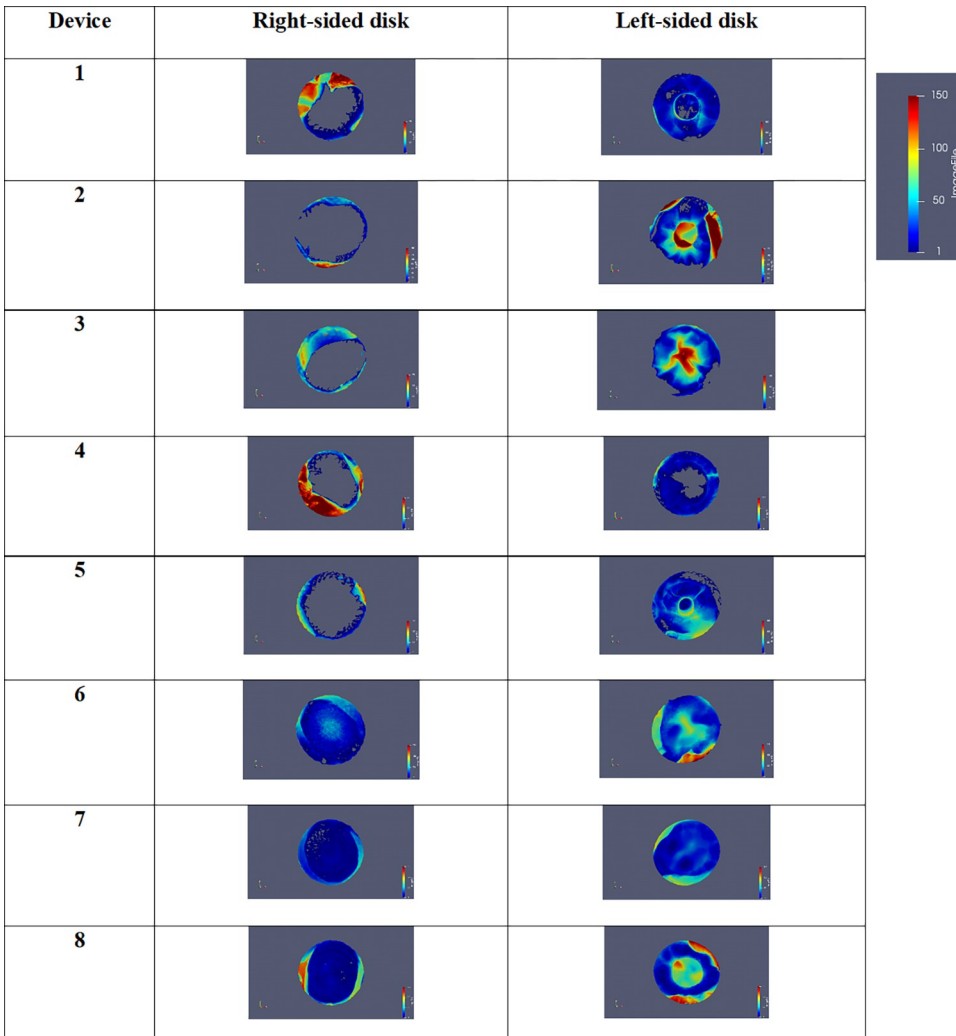

**Fig 9. Thickness coverage of the disks of the eight devices.** Devices 1–3 explanted after one-month follow-up. Devices 4–8 explanted after 3-month follow-up. The color scale was the same for all pictures (blue: Low thickness, red: High thickness).

Currently, non-invasive imaging techniques such as MRI and CT-scan have insufficient resolution and are susceptible to image artefacts because of the metal framework of the device [32]. Our study provides a methodology for preclinical evaluation and robust device testing with resolution requirements of CT imaging to characterize tissue coverage of the occluders. Developments in clinical imaging systems remain challenging to meet these requirements for application in patient care. The highlighted determinants for successful tissue coverage are namely complete deployment of the occluder.

Yet, the application of micro-CT imaging to measure device coverage provides a first step towards determining the distinction between thrombus and physiological healing process, as shown in left atrial appendage occluder devices [33].

## Study limitations

This study was performed in a pre-clinical setting, within a small population of sheep and applied to a single device model. The use of micro-CT, an in vitro imaging technique, was

tested, showing its potential for identifying the tissue coverage of an ASD occluder device. Other pre-clinical experiments, using in vivo CT-scanning, should be performed to determine how to identify the tissue coverage on intra-cardiac devices and be able to translate it to clinical practice.

## Conclusion

The assessment of tissue reactions, including endothelialisation, after implantation of an ASD occluder can rely on in vitro micro-CT analysis, which is complementary to histology.

Translation of this methodology to clinical practice remains challenging due to the demand for high resolution in beating heart conditions. However, it would allow an individual follow-up, to avoid complications related to thrombi and infections.

## Supporting information

**S1 File. Data of the eight animals included in the study.**
(XLSX)

**S2 File. Macroscopic examination of the disks of the eight devices, explanted at one month (1 to 3) or 3 months (4 to 8).**
(TIF)

## Author Contributions

**Conceptualization:** Elodie Perdreau, Zakaria Jalal, Matthias Sigler, Hubert Cochet, Jérôme Naulin, Bruno Quesson.

**Data curation:** Elodie Perdreau, Richard D. Walton, Matthias Sigler, Jérôme Naulin.

**Formal analysis:** Elodie Perdreau, Richard D. Walton, Matthias Sigler, Jérôme Naulin.

**Funding acquisition:** Elodie Perdreau, Jean-Benoît Thambo.

**Investigation:** Elodie Perdreau, Zakaria Jalal, Richard D. Walton, Matthias Sigler.

**Methodology:** Elodie Perdreau, Zakaria Jalal, Richard D. Walton, Matthias Sigler, Hubert Cochet, Jérôme Naulin, Bruno Quesson.

**Project administration:** Olivier Bernus, Jean-Benoît Thambo.

**Resources:** Olivier Bernus, Jean-Benoît Thambo.

**Software:** Richard D. Walton.

**Supervision:** Zakaria Jalal, Matthias Sigler, Bruno Quesson, Jean-Benoît Thambo.

**Validation:** Zakaria Jalal, Richard D. Walton, Matthias Sigler.

**Writing – original draft:** Elodie Perdreau, Richard D. Walton.

**Writing – review & editing:** Zakaria Jalal, Matthias Sigler, Hubert Cochet, Jérôme Naulin, Bruno Quesson, Olivier Bernus, Jean-Benoît Thambo.

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
