## [Decision Letter · Decision Letter 0]

31 Oct 2022

PONE-D-22-27207Assessment of cardiac percutaneous occluders healing process using computed tomography imagingPLOS ONE

Dear Dr. Perdreau,

Thank you for submitting your manuscript to PLOS ONE. After careful consideration, we feel that it has merit but does not fully meet PLOS ONE’s publication criteria as it currently stands. Therefore, we invite you to submit a revised version of the manuscript that addresses the points raised during the review process.

We look forward to receiving your revised manuscript.

Kind regards,

Chengming Fan, MD, PhD

Academic Editor

PLOS ONE

Journal Requirements:

2. To comply with PLOS ONE submissions requirements, in your Methods section, please provide additional information regarding the experiments involving animals and ensure you have included details on all efforts to alleviate suffering.

3. We note that Figure 1 in your submission contain copyrighted images. All PLOS content is published under the Creative Commons Attribution License (CC BY 4.0), which means that the manuscript, images, and Supporting Information files will be freely available online, and any third party is permitted to access, download, copy, distribute, and use these materials in any way, even commercially, with proper attribution. For more information, see our copyright guidelines: http://journals.plos.org/plosone/s/licenses-and-copyright.

Additional Editor Comments:

Please response to the reviewers point by point

Reviewers' comments:

Reviewer's Responses to Questions

**Comments to the Author**

1. Is the manuscript technically sound, and do the data support the conclusions?

Reviewer #1: Yes

Reviewer #2: Yes

2. Has the statistical analysis been performed appropriately and rigorously? 

Reviewer #1: N/A

Reviewer #2: I Don't Know

3. Have the authors made all data underlying the findings in their manuscript fully available?

Reviewer #1: No

Reviewer #2: Yes

4. Is the manuscript presented in an intelligible fashion and written in standard English?

Reviewer #1: Yes

Reviewer #2: Yes

5. Review Comments to the Author

Reviewer #1: This article by Dr. Perdreau and colleagues from Bordeaux University Hospital reported the assessment of healing process of implanted ASD devices using micro-CT and histological analyses. In general, the idea is unique and interesting. These data are also valuable and important for all the scientists and physicians working on ASD device development.

For better understanding of healing process, immunohistochemical staining of ET cells would be necessary.

I have several comments and suggestions.

1. In abstract, and in line 264-265, clarify the follow up periods in the data of surface coverage of 93 % for LA side and 55% for RA side. Are these data from all 8 animals?

As there are two groups with different follow up periods, it should be separately described.

2. Describe the possible reason why LA side had higher coverage in discussion.

3. Histological analysis should include endothelial cells specific staining.

4. There was no figure legends. Provide it.

5. Describe the success rate of Brockenbrough technique and if there was any support image (like TEE or ICE) for this?

Reviewer #2: The title does not meet study design/ composition. Micro CT was used. It should be mentioned. Also only ASD device: Nit occlud was studied. Results should not generalised for all devices.

Altough the aim of the study is to evaluate device coverage process, the feasibility of CT scan for this purpose, authors studied this only after heart explant. The CT scans performed before heart explant and a second CT scan after explantation process should be beneficial for comparing the results of images with beating and non-beating heart. If the aim was evaluting the epitalization status CT scan is not necessary we can see macroscopically and microscopically. If they want to evalute the feasibility of CT scan than it was necessary to evalute it on beating heart.

We know that when we make a cardiac scan with CT on beating heart with high rates results has many image artefacts. Was that the fact that they perform the study after explantation?

So what is the clinical and practical advantage of this study? Discussion must focus on this subject.

6. PLOS authors have the option to publish the peer review history of their article (what does this mean?). If published, this will include your full peer review and any attached files.

Reviewer #1: No

Reviewer #2: No

---

## [Author Response · Author response to Decision Letter 0]

15 Jan 2023

Journal Requirements:

=> We have taken into consideration this remark.

2. To comply with PLOS ONE submissions requirements, in your Methods section, please provide additional information regarding the experiments involving animals and ensure you have included details on all efforts to alleviate suffering.

=> We provided additional information regarding the experiments involving animals and ensured to have included details on all efforts to alleviate suffering.

3. We note that Figure 1 in your submission contain copyrighted images. All PLOS content is published under the Creative Commons Attribution License (CC BY 4.0), which means that the manuscript, images, and Supporting Information files will be freely available online, and any third party is permitted to access, download, copy, distribute, and use these materials in any way, even commercially, with proper attribution. For more information, see our copyright guidelines: http://journals.plos.org/plosone/s/licenses-and-copyright.

=> We decided to remove the figure under copyright from the submission. 

Additional Editor Comments:

Please response to the reviewers point by point

Reviewers' comments:

Reviewer's Responses to Questions

Comments to the Author

1. Is the manuscript technically sound, and do the data support the conclusions?

Reviewer #1: Yes

Reviewer #2: Yes

2. Has the statistical analysis been performed appropriately and rigorously? 

Reviewer #1: N/A

Reviewer #2: I Don't Know

3. Have the authors made all data underlying the findings in their manuscript fully available?

Reviewer #1: No

Reviewer #2: Yes

4. Is the manuscript presented in an intelligible fashion and written in standard English?

Reviewer #1: Yes

Reviewer #2: Yes

5. Review Comments to the Author

Reviewer #1: This article by Dr. Perdreau and colleagues from Bordeaux University Hospital reported the assessment of healing process of implanted ASD devices using micro-CT and histological analyses. In general, the idea is unique and interesting. These data are also valuable and important for all the scientists and physicians working on ASD device development.

For better understanding of healing process, immunohistochemical staining of ET cells would be necessary.

I have several comments and suggestions.

1. In abstract, and in line 264-265, clarify the follow up periods in the data of surface coverage of 93 % for LA side and 55% for RA side. Are these data from all 8 animals?

As there are two groups with different follow up periods, it should be separately described.

=> The median of surface coverage was calculated for the 8 cases. We also provided the calculation for the two groups. 

2. Describe the possible reason why LA side had higher coverage in discussion.

=> The complete deployment of the occluder showed improved coverage but also a higher propensity for total transoccluder tissue replacement. The mechanisms promoting elevated tissue replacement remains to be investigated. These may include the mechanical status of the surrounding tissue, smaller occlusion diameters and reduced transoccluder distances and the efficiency of septal occlusion. 

3. Histological analysis should include endothelial cells specific staining.

=>The goal of histology was to identify the device coverage. Richardson blue staining allows for definite identification of endothelial cells based on morphological criteria. Thus, we decided not to perform endothelial cells specific staining. Here is a highly zoomed image of the histological analysis where we can recognize endothelial cells (outer layer at the superior part of the picture and around the metal struts).

4. There was no figure legends. Provide it.

=> As recommended by the submission guidelines, we provided the figure legends directly in the manuscript.

5. Describe the success rate of Brockenbrough technique and if there was any support image (like TEE or ICE) for this?

=> The success rate of Brockenbrough technique was 100%. We added a TEE image to support it. 

Reviewer #2: The title does not meet study design/ composition. Micro CT was used. It should be mentioned. Also only ASD device: Nit occlud was studied. Results should not generalised for all devices.

=> We changed the title, taking into consideration your recommendations.

Altough the aim of the study is to evaluate device coverage process, the feasibility of CT scan for this purpose, authors studied this only after heart explant. The CT scans performed before heart explant and a second CT scan after explantation process should be beneficial for comparing the results of images with beating and non-beating heart. If the aim was evaluting the epitalization status CT scan is not necessary we can see macroscopically and microscopically. If they want to evalute the feasibility of CT scan than it was necessary to evalute it on beating heart.

We know that when we make a cardiac scan with CT on beating heart with high rates results has many image artefacts. Was that the fact that they perform the study after explantation?

=> We are addressing a need to better evaluate the coverage of devices in a preclinical setting. We found that micro-CT robustly characterizes the full 3D aspect of the device and its coverage, that no other imaging tool can effectively achieve in the same time frame. A beating heart system would aid understanding how this type of evaluation would perform in clinical applications, but the tools and resolution on the systems are not currently available. Despite this, CT and micro-CT systems are capable of ECG-gated imaging acquisition. Therefore, the potential for beating heart imaging is already evident for future studies and clinical applications. 

So what is the clinical and practical advantage of this study? Discussion must focus on this subject.

=> Our study provides the resolution requirements of CT imaging to characterize tissue coverage of the occluders. Developments in clinical imaging systems remain to be put in place to meet these requirements for application in patient care. However, the study provides a methodology for preclinical evaluation and robust device testing. The determinants for successful tissue coverage are namely complete deployment of the occluder. This study highlights the main factors to verify during procedures to improve treatment occlusion efficiency. We included this question in the discussion of the manuscript. 

6. PLOS authors have the option to publish the peer review history of their article (what does this mean?). If published, this will include your full peer review and any attached files.

Do you want your identity to be public for this peer review? For information about this choice, including consent withdrawal, please see our Privacy Policy.

Reviewer #1: No

Reviewer #2: No

---

## [Decision Letter · Decision Letter 1]

27 Jan 2023

PONE-D-22-27207R1Assessment of Nit-Occlud atrial septal defect occluder device healing process using micro-computed tomography imagingPLOS ONE

Dear Dr. Perdreau,

Thank you for submitting your manuscript to PLOS ONE. After careful consideration, we feel that it has merit but does not fully meet PLOS ONE’s publication criteria as it currently stands. Therefore, we invite you to submit a revised version of the manuscript that addresses the points raised during the review process.

We look forward to receiving your revised manuscript.

Kind regards,

Chengming Fan, MD, PhD

Academic Editor

PLOS ONE

Reviewers' comments:

Reviewer's Responses to Questions

**Comments to the Author**

1. If the authors have adequately addressed your comments raised in a previous round of review and you feel that this manuscript is now acceptable for publication, you may indicate that here to bypass the “Comments to the Author” section, enter your conflict of interest statement in the “Confidential to Editor” section, and submit your "Accept" recommendation.

Reviewer #1: All comments have been addressed

Reviewer #3: All comments have been addressed

2. Is the manuscript technically sound, and do the data support the conclusions?

Reviewer #1: Partly

Reviewer #3: Partly

3. Has the statistical analysis been performed appropriately and rigorously? 

Reviewer #1: N/A

Reviewer #3: No

4. Have the authors made all data underlying the findings in their manuscript fully available?

Reviewer #1: Yes

Reviewer #3: No

5. Is the manuscript presented in an intelligible fashion and written in standard English?

Reviewer #1: Yes

Reviewer #3: Yes

6. Review Comments to the Author

Reviewer #1: I think that the authors appropriately answered all of the questions by the reviewers. And the revision was appropriately done and the revised manuscript is much improved.

Reviewer #3: First, I would like to congratulate Dr Perdreau et al et al for their work in using a sheep model and trying to evaluate its endothelialization process with Micro CT. It is hoped that such minimally invasive imaging evaluation methods will be applied clinically in the future.

I have a few questions and points to make.

1) Why did this group use the sheep model instead of pigs, etc.? An explanation is needed.

2) Why did you use PFM medical out of all the ASD devices?　Explanation is needed.

3) Did you acquire ICE or TEE echo-images during the surgery?　When experimenting with ASD devices in sheep, it should be impossible to place them in the correct position without ICE or other guiding.　And even with echo, you need to follow up to see if the shunt, etc. is closed and the implanted device is in the correct position.　

4) Dr Perdreau et al. have animals sacrificed at 1 month and 3 months, but the observation period is too short. If endothelialization is to be considered, shouldn't evaluation at 6 months and 1 year, etc. be necessary?　

5) If endothelialization is to be evaluated, immunostaining with CD 31,vWF seems necessary.　

6) Why is endothelialization on the right atrial side slower than on the left atrial side? This should be added to the discussion.

7. PLOS authors have the option to publish the peer review history of their article (what does this mean?). If published, this will include your full peer review and any attached files.

Reviewer #1: No

Reviewer #3: No

---

## [Author Response · Author response to Decision Letter 1]

15 Mar 2023

Thank you for reviewing our work. Here is the answers to Reviewer 3 comments.

Reviewer #1: I think that the authors appropriately answered all of the questions by the reviewers. And the revision was appropriately done and the revised manuscript is much improved.

We thank this reviewer for the constructive comments and revision. 

Reviewer #3: First, I would like to congratulate Dr Perdreau et al et al for their work in using a sheep model and trying to evaluate its endothelialization process with Micro CT. It is hoped that such minimally invasive imaging evaluation methods will be applied clinically in the future.

I have a few questions and points to make.

We thank you for reviewing our work. We carefully answered the questions (see below). 

1) Why did this group use the sheep model instead of pigs, etc.? An explanation is needed.

We chose a sheep model instead of pig model because a part of the inital experimental protocol included MRI, so we needed an animal that was not going to put on weight too much. The healing process was also close to human being. 

2) Why did you use PFM medical out of all the ASD devices? Explanation is needed.

A reduced metal content and absence of protruding fixation clamps improved the compatibility of the Nit-Occlud ASD-R occluder with computed tomography-based imaging techniques, which are prone to artefacts associated with over attenuation of X-rays. Before this experimental protocol, we notably experimented the use of Amplatzer septal occluder but the protruding fixation resulted in important artefacts when performing CT scan.

3) Did you acquire ICE or TEE echo-images during the surgery? When experimenting with ASD devices in sheep, it should be impossible to place them in the correct position without ICE or other guiding. And even with echo, you need to follow up to see if the shunt, etc. is closed and the implanted device is in the correct position.

We used TTE to guide the procedure (with fluoroscpic guidance associated) and acquired echo-images during the procedure : TTE allowed to follow the wire to perform trans-septal puncture, see the absence of atrial shunt at the end of the procedure and the correct position of the device. We included an example of acquired echo-image in Figure 1. Before sacrifice, we performed TTE to check the absence of residual shunt and confirm good position of the device. We manage to obtain images of enough quality and didn’t need to use TEE. 

4) Dr Perdreau et al. have animals sacrificed at 1 month and 3 months, but the observation period is too short. If endothelialization is to be considered, shouldn't evaluation at 6 months and 1 year, etc. be necessary?　

We chose this period of observation, regarding the duration found in literature (28-90 days), which is quicker than in humans. We can also see that endothelialization of the device was complete after 3 months for devices without any protrusion of the disk. Having incomplete endothelialization of some devices allowed to be sure to identify the covering tissue. 

5) If endothelialization is to be evaluated, immunostaining with CD 31,vWF seems necessary.　

The goal of histology was to identify the device coverage. Richardson blue staining allows for definite identification of endothelial cells based on morphological criteria. Thus, we decided not to perform endothelial cells specific staining. Here is a highly zoomed image of the histological analysis where we can recognize endothelial cells (outer layer at the superior part of the picture and around the metal struts).

6) Why is endothelialization on the right atrial side slower than on the left atrial side? This should be added to the discussion.

Disk conformation appears to determine, in part, device coverage. The complete deployment of the occluder showed improved coverage but also a higher propensity for total transoccluder tissue replacement. Complete transoccluder tissue replacement in fully deployed occluders appeared to facilitate tissue coverage of the right disk. The mechanisms promoting elevated tissue replacement remains to be investigated. These may include the mechanical status of the surrounding tissue, smaller occlusion diameters, reduced transoccluder distances and the efficiency of septal occlusion. This was added in the discussion.

---

## [Decision Letter · Decision Letter 2]

3 Apr 2023

Assessment of Nit-Occlud atrial septal defect occluder device healing process using micro-computed tomography imaging

PONE-D-22-27207R2

Dear Dr. Perdreau,

We’re pleased to inform you that your manuscript has been judged scientifically suitable for publication and will be formally accepted for publication once it meets all outstanding technical requirements.

Kind regards,

Chengming Fan, MD, PhD

Academic Editor

PLOS ONE

Additional Editor Comments (optional):

Reviewers' comments:

Reviewer's Responses to Questions

**Comments to the Author**

1. If the authors have adequately addressed your comments raised in a previous round of review and you feel that this manuscript is now acceptable for publication, you may indicate that here to bypass the “Comments to the Author” section, enter your conflict of interest statement in the “Confidential to Editor” section, and submit your "Accept" recommendation.

Reviewer #1: All comments have been addressed

Reviewer #3: All comments have been addressed

2. Is the manuscript technically sound, and do the data support the conclusions?

Reviewer #1: Yes

Reviewer #3: Yes

3. Has the statistical analysis been performed appropriately and rigorously? 

Reviewer #1: Yes

Reviewer #3: Yes

4. Have the authors made all data underlying the findings in their manuscript fully available?

Reviewer #1: Yes

Reviewer #3: Yes

5. Is the manuscript presented in an intelligible fashion and written in standard English?

Reviewer #1: Yes

Reviewer #3: Yes

6. Review Comments to the Author

Reviewer #1: I think that the authors appropriately answered all of the questions by the reviewer. And the revision was appropriately done and the revised manuscript is much improved.

Reviewer #3: Again ,I would like to congratulate Dr Perdreau et al et al for their work in

using a sheep model and trying to evaluate its endothelialization process with Micro

CT. Their revison looks reasonable for me.

7. PLOS authors have the option to publish the peer review history of their article (what does this mean?). If published, this will include your full peer review and any attached files.

Reviewer #1: No

Reviewer #3: No

---

## [Editor Report · Acceptance letter]

12 Apr 2023

PONE-D-22-27207R2 

Assessment of Nit-Occlud atrial septal defect occluder device healing process using micro-computed tomography imaging 

Dear Dr. Perdreau:

I'm pleased to inform you that your manuscript has been deemed suitable for publication in PLOS ONE. Congratulations! Your manuscript is now with our production department. 

Kind regards, 

on behalf of

Dr. Chengming Fan 

Academic Editor

PLOS ONE